# Sorption of Monothioarsenate to the Natural Sediments and Its Competition with Arsenite and Arsenate

**DOI:** 10.3390/ijerph182312839

**Published:** 2021-12-06

**Authors:** Huimei Shan, Jinxian Zhang, Sanxi Peng, Hongbin Zhan, Danxue Liao

**Affiliations:** 1Guangxi Key Laboratory of Environmental Pollution Control Theory and Technology, Guilin University of Technology, Guilin 541004, China; groundwater_zhang@163.com (J.Z.); groundwater_liao@163.com (D.L.); 2College of Environmental Science and Engineering, Guilin University of Technology, Guilin 541004, China; 3College of Earth Science, Guilin University of Technology, Guilin 541004, China; 2016047@glut.edu.cn; 4Department of Geology & Geophysics, Texas A&M University, College Station, TX 77843, USA

**Keywords:** monothioarsenate, natural sediment, adsorption, competitive effect

## Abstract

Monothioarsenate (MTAs^V^) is one of the major arsenic species in sulfur- or iron-rich groundwater, and the sediment adsorption of MTAs^V^ plays an important role in arsenic cycling in the subsurface environment. In this study, batch experiments and characterization are conducted to investigate the sorption characteristic and mechanism of MTAs^V^ on natural sediments and the influences of arsenite and arsenate. Results show that MTAs^V^ adsorption on natural sediments is similar to arsenate and arsenite, manifested by a rapid early increasing stage, a slowly increasing stage at an intermediate time until 8 h, before finally approaching an asymptote. The sediment sorption for MTAs^V^ mainly occurs on localized sites with high contents of Fe and Al, where MTAs^V^ forms a monolayer on the surface of natural sediments via a chemisorption mechanism and meanwhile the adsorbed MTAs^V^ mainly transforms into other As species, such as AlAs, Al-As-O, and Fe-As-O compounds. At low concentration, MTAs^V^ sorption isotherm by natural sediments becomes the Freundlich isotherm model, while at high concentration of MTAs^V^, its sorption isotherm becomes the Langmuir isotherm model. The best-fitted maximum adsorption capacity for MTAs^V^ adsorption is about 362.22 μg/g. Furthermore, there is a competitive effect between MTAs^V^ and arsenate adsorption, and MTAs^V^ and arsenite adsorption on natural sediments. More specifically, the presence of arsenite greatly decreases MTAs^V^ sorption, while the presence of MTAs^V^ causes a certain degree of reduction of arsenite adsorption on the sediments before 4 h, and this effect becomes weaker when approaching the equilibrium state. The presence of arsenate greatly decreases MTAs^V^ sorption and the presence of MTAs^V^ also greatly decreases arsenate sorption. These competitive effects may greatly affect MTAs^V^ transport in groundwater systems and need more attention in the future.

## 1. Introduction

Arsenic (As) pollution has become a global environmental concern and long-term intake of high-arsenic groundwater is threatening human health seriously [1,2]. Commonly, in an oxidizing environment, arsenate of H_2_AsO_4_^−^ is the dominant arsenic species under pH of lower than 6.9 [3], but in a reducing environment, the dominant arsenic species become arsenite of H_3_AsO_3_^0^ at a pH of lower than 9.2, and arsenite can transform into AsO_4_^3−^ at pH of higher than 9.2 [3,4]. However, when sulfur or insoluble sulfide coexists with arsenite or arsenate, oxygen-bonded arsenic will be substituted by sulfur to form As-SH and/or As=S substructures, which is named as thioarsenic, commonly including thioarsenite (TAs^III^) and thioarsenate (TAs^V^) [5,6,7]. According to the extent of S-substitution, TAs^III^ is further divided into mono-, di-, and tri-thioarsenite (H_n_As^III^O_3−x_S_x_^n−3^ with x = 1~3 and *n* = 0~3) and TAs^V^ is further divided into mono-, di-, tri-, and tetra-thioarsenate (H_n_As^V^O_4-x_S_x_^n−3^ with x = 1–4 and *n* = 0–3), respectively [8]. Recently, many field investigations show that TAs^V^ is the dominant As species rather than arsenate in sulfur-rich groundwater [9,10,11]. The percentage of TAs^V^ in total arsenic can account for as much as 80% in some geothermal water samples [12] and 68% in fresh groundwater samples [13]. Furthermore, TAs is also observed in the rice paddy pore waters [14,15], and a few studies on the toxicity of TAs^V^ show that the toxicity of monothioarsenate is similar to that of arsenate but lower than that of arsenite [16]. Based on these findings, studies on the distribution, characteristics, and fate of thioarsenic species in the aquatic environment are attracting increasing attention in recent decades.

Sorption/desorption is one of the most important mechanisms controlling the behavior of As in the natural environment [17,18,19]. In the past decades, the sorption of arsenite and arsenate by various mediums (especially Fe minerals) has been extensively studied [20,21,22,23,24,25,26]. Compared to arsenite and arsenate, thioarsenic adsorption is much less known. Only a few researchers have reported that the sorption of thioarsenic is significantly different from that of arsenite and/or arsenate even on the same adsorbents. For example, Couture et al. (2013) have found that monothioarsenate (MTAs^V^), tetra-thioarsenate (TTAs^V^), arsenite, and arsenate show significantly different sorption affinities on iron minerals including 2-line ferrihydrite, goethite, mackinawite, and pyrite [27]. MTAs^V^ adsorption capacity on goethite is lower than arsenite and arsenate, and its sorption kinetic is slower than arsenite but is comparable with arsenate [28]. For mackinawite, the adsorption of MTAs^V^ and TTAs^V^ is lower than arsenite and arsenate, but for pyrite, the order of adsorption from highest to lowest is MTAs^V^, arsenite, arsenate, and TTAs^V^ [27]. Recent studies have reported that TAs^III^ shows smaller adsorption and weaker affinity on magnetite, ferrous sulfide, and hematite compared to arsenite, and the formation of TAs^III^ can greatly inhibit these three iron minerals sorption for arsenic at a pH of 7–11 [29,30]. Similar to iron minerals, the aluminum-bearing mineral of amorphous Al(OH)_3_ sorbs much less As from thioarsenic solutions than that from arsenate solutions. But it is worth noticing that MTAs^V^ is partly reduced to arsenite in an aqueous solution during the adsorption process and the acid-catalysis at the surface of amorphous Al(OH)_3_ can enhance this transformation, especially in the acid environment [31]. Until now, most studies about thioarsenic sorption are mainly related to pure Fe- or Al-bearing minerals. At present, to our best knowledge, there is no study focusing on thioarsenic sorption on natural sediments. Furthermore, MTAs^V^ is not only one of the major thioarsenic species in natural sulfidic waters, but it also serves as an important intermediate product during thioarsenic transformation to arsenite or arsenate [32], however, it is still unknown about MTAs^V^ sorption to natural sediments and the influence of various hydrogeochemical factors on MTAs^V^ sorption.

This study is aimed to investigate the characteristics of MTAs^V^ sorption to natural sediments. Characterization analysis including structure, surface morphology, surface elements composition, surface area, and pore volume of sediment is also conducted to further identify the primary mechanisms of MTAs^V^ sorption to natural sediments. Furthermore, batch experiments are carried out to further determine the influences of arsenite and arsenate on MTAs^V^ sorption.

## 2. Experiments and Methods

### 2.1. Materials

The standard MTAs^V^ sample is synthesized based on the method proposed by Keller et al. [9]. 5.0 g As_2_O_3_ (>99.5%) and 6.0 g NaOH (>96%) are dissolved into 20 mL fresh oxygen-free water, and then 1.44 g sulfur (>99%) is added into the mixture solution to heat at 100 °C for 2 h. After that, the excess sulfur is filtered out. The residual solution is cooled at 4 °C for 3–5 min and then stored in an anaerobic glove chamber until an acicular crystal is formed. This crystal is dissolved into 10 mL fresh oxygen-free water and then recrystallized by adding 5 mL anhydrous ethanol to obtain the standard MTAs^V^. The purity is determined to be 98% using the Liquid Chromatography Hydride-Generation Atomic Fluorescence (LC-HG-AFS) method [33]. The major component is determined to be Na_3_AsSO_3_·7H_2_O and impurity is arsenite using the X-ray powder diffraction (XRD) (X’Pert3, Netherlands, Cu target, λ = 1.54056 Å).

Standard arsenite and arsenate samples are purchased from China Academy of Metrology and their molarities are (1.011 ± 0.016) μmol/g and (0.233 ± 0.005) μmol/g, respectively. Other chemicals including Na_2_HAsO_4_·7H_2_O, NaAsO_2_, and KCl are analytical reagents and purchased from Xilong Science Co., LTD of China.

The sediment samples are collected from 7.8–8.2 m depth in Xiantao city, Hubei Province, Jianghan Basin, China, where the long-term field monitoring investigations have found that about 87% of the groundwater samples in this area have As concentration exceeding the WHO guideline limit value of 10 μg/L, up to 2328 μg/L, which primarily occurred between depths of 10 and 45 m [34]. Our recent study of hydrochemical modeling in this area has found that MTAs^V^ may account for about 56.82–94.50% of total As in groundwater. After collection, samples are capped with PTFE lids and wax-sealed immediately, and then they are stored in an opaque anaerobic box at 4 °C. Before adsorption experiment, the sediment is naturally dried and sieved to obtain samples with particle sizes of 0.150–0.180 mm. In this study, to avoid the effects of As contents on the MTAs^V^ adsorption, the natural sediment samples without As contamination are used for laboratory experiments. Groundwater solutions are synthesized according to the field investigation results of groundwater in this area [34]. The hydrochemical parameters of synthesized groundwater (SGW) solutions are shown in Table 1. The SGW solutions are used to prepare MTAs^V^ solutions for the following batch experiments.

### 2.2. Batch Experiments

In this case, 0.50 g sediment samples are added into a series of 50 mL SGW solutions with initial MTAs^V^ concentrations ranging from 0.002 to 0.667 mmol/L (0.15 mg As/L to 50.00 mg As/L). After reaction for 12 h, the supernatant is collected for As species and concentration analysis at prescribed time intervals. To avoid the possible oxidation or decomposition of MTAs^V^, all the reaction tubes are full of the experimental solutions without leaving any headspace. Meanwhile, the blank experiments without sediment samples addition are conducted to investigate MTAs^V^ stability within 24 h.

To investigate the mutual interferences between MTAs^V^ and other As species, the same concentrations of arsenite and arsenate are added into a series of mixtures containing 0.50 g sediments and 50 mL 0.627 mmol/L MTAs^V^ solutions, respectively. During the reaction of 12 h, the supernatants are collected at prescribed time intervals and used for As species and concentration analysis.

All the above experiments are carried out twice at the same temperature of about 25 °C. All the aqueous samples are filtered by 0.45 μm filter before As analysis.

The sorption capacity on natural sediments was calculated using the following equations:(1)$$Q_{t}=\frac{V_{s}(C_{0}-C_{t})}{m_{s}}$$
(2)$$Q_{e}=\frac{V_{s}(C_{0}-C_{e})}{m_{s}}$$
where *Q_t_* and *Q_e_* (μg/g) are sorption capacities at time *t* and equilibrium stage, respectively. vs. (mL) is the volume of groundwater solution, *C*_0_ (mg/L) is the initial concentration, *C_t_* and *C_e_* (mg/L) are the concentrations at time *t* and equilibrium stage, respectively. *m_s_* (g) is the mass of adsorbent.

### 2.3. Kinetic and Isotherm Models

The kinetic adsorption results of MTAs^V^ on the sediment are used to fit with five models including pseudo-first-order, pseudo-second-order, elovich, intraparticle diffusion, and fractional power models, which have been widely applied to characterize the sorption processes governing the behavior of different adsorbates [35,36,37,38,39,40]. Their expression equations are shown as Equations (3)–(7), respectively.
(3)$$Q_{t}=Q_{e}\left(1-e^{-k_{1}t}\right)$$
(4)$$Q_{t}=\frac{Q_{e}^{2}k_{2}t}{1+Q_{e}k_{2}t}$$
(5)$$Q_{t}=\frac{1}{\beta }\cdot \ln\left(\alpha \beta \right)+\frac{1}{\beta }\ln\left(t\right)$$
(6)$$Q_{t}=k_{i}{\left(t\right)}^{0.5}+c$$
(7)$$\mathrm{In}(Q_{t})=\ln\left(a\right)+b\ln\left(t\right)$$
where *Q_e_* and *Q_t_* (μg/g) are the adsorption capacities at equilibrium state and time *t*, respectively; *t* (h) is the adsorption time; *k*_1_ (h^−1^) and *k*_2_ (g/(μg·h)) are the pseudo-first-order and pseudo-second-order rate sorption constants, respectively; α is the rate of chemisorption at zero coverage (μg/(g·h)) and *β* is the extent of surface coverage and activation energy for chemisorption (g/μg); *k_i_* (h^−1^) means the intra-particle diffusion rate constant; *a, b* and *c* are constants. MTAs^V^ adsorption isotherm on natural sediments are used to fit with the Langmuir, Freundlich, and Langmuir-Freundlich models. Their equations are as follows:

Langmuir model:(8)$$Q_{e}=\frac{Q_{m}K_{L}C_{e}}{1+K_{L}C_{e}}$$

Freundlich model:(9)$$Q_{e}=K_{F}C_{e}^{1/n}$$

Langmuir-Freundlich model:(10)$$Q_{e}=\frac{Q_{m}{\left(K_{LF}C_{e}\right)}^{m}}{1+{\left(K_{LF}C_{e}\right)}^{m}}$$
where *Q_e_* is the adsorption capacity of MTAs^V^ at equilibrium stage (μg/g); *Q_m_* is the maximum equilibrium adsorption capacity of MTAs^V^ (μg/g); *C_e_* is the concentration of MTAs^V^ at equilibrium stage (mg/L); *K_L_* and *K_F_* are adsorption constants of Langmuir and Freundlich models, respectively; *K_LF_* is the equilibrium constant for heterogeneous solids; 1/*n* is the adsorption strength constant of the Freundlich equation, and a smaller 1/*n* means a better adsorption capacity; *m* is a heterogeneous parameter and it lies between 0 and 1.

### 2.4. Measurement

The concentrations of As species including MTAs^V^, arsenite, and arsenate are determined using Liquid Chromatography Hydride-Generation Atomic Fluorescence (LC-HG-AFS) according to our previously reported method [33]. The pH value is measured using a Hach-HQ30d analyzer. The crystal structure of sediment is determined by X-ray powder diffraction (XRD) (X’Pert3, Netherlands, Cu target, λ = 1.54056 Å). The scanning step, speed, and range are 0.0263°, 0.6565°/s, and 5°–90° (2θ), respectively. Surface morphology and elemental analysis of sediment are determined using the Scanning Electron Microscope (SEM) and Energy Dispersion Spectrum (EDS) (JSM-7900F, JEOL, Tokyo, Japan). Surface area and pore volume are determined by the adsorption method using surface area and porosity analyzer (Gaobo, China). The sediment sample is dried at 200 °C before measurement, the working temperature is 77K, and the sorbent is nitrogen gas.

## 3. Results and Analysis

### 3.1. MTAs^V^ Stability

In the blank experiments, the ratio of MTAs^V^ concentration at time *t* (*C_t_*) and the initial concentration (*C*_0_) over time is shown in Figure 1. It is found that the *C_t_*/*C*_0_ values basically equal 1 and show almost no changes within 24 h, indicating that MTAs^V^ is stable and does not transform into other As species during the experiments in this study. Furthermore, in the batch adsorption experiments, no As species production except the added MTAs^V^ are detected in the solutions within 24 h using LC-HG-AFS, meaning that MTAs^V^ can remain stable during its adsorption.

### 3.2. Sorption Kinetics of MTAs^V^ by the Sediments

The adsorption kinetics of 0.002, 0.013, and 0.667 mmol/L MTAs^V^ on the natural sediment are shown in Figure 2. Overall, the adsorption capacity of MTAs^V^ (*Q_t_*) increases rapidly in a short time and then increases slowly before 8 h, and finally approaches an asymptote. This is consistent with the previous studies about the sorption of arsenite, arsenate, and thioarsenic on minerals and soils [19,27,41,42]. Specifically, when the concentration of MTAs^V^ is as low as 0.002 mmol/L (which is equivalent to 0.15 mg As/L) (Figure 2a), *Q_t_* increases rapidly to 7.09 μg/g within 10 min. After that, it increases slowly to 9.64 μg/g before 8 h and finally tends to be stable after that. For the mid-concentration of 0.013 mmol/L MTAs^V^ (which is equivalent to 1.00 mg As/L) (Figure 2b), it is found that *Q_t_* increases rapidly to 37.54 μg/g within 30 min, then increases slowly to 42.90 μg/g at 8 h, and finally tends to be stable after that. For the high concentration of 0.667 mmol/L MTAs^V^ (which is equivalent to 50.00 mg As/L) (Figure 2c), *Q_t_* increases rapidly to 181.00 μg/g within 50 min, then increases slowly to 219.50 μg/g before 8 h, and finally keeps stable after that. In the initial stage, *Q_t_* increases rapidly because there are enough reactive sites on the surface of the sediment to adsorb a large number of MTAs^V^ [33]. This stage becomes longer with the increasing concentration of MTAs^V^, which may be because it needs more time for the higher concentration of MTAs^V^ to be adsorbed by the solid phase. After that, MTAs^V^ adsorption becomes slower, and the adsorption rate decreases, which may be because internal diffusion becomes the dominant process in this stage, and the internal diffusion resistance increases with more MTAs^V^ adsorbed by the solid phase [33]. From 8 h to 12 h, all the adsorption of different concentrations of MTAs^V^ reaches equilibrium when the internal diffusion resistance is very high and the mass concentration driving force is too low to force MTAs^V^ to be adsorbed by the solid phase. Therefore, considering the stability of MTAs^V^, 8 h is determined to be the adsorption equilibrium time for different concentrations of MTAs^V^ adsorbed by natural sediments.

Figure 2 and Table 2 show the best fitting results of MTAs^V^ sorption to natural sediments using the pseudo-first-order and the pseudo-second-order kinetic models. For MTAs^V^ concentrations of 0.002 and 0.013 mmol/L, their adsorption experimental data fit better into the pseudo-second-order kinetic model with a coefficient of determination (*R*^2^) greater than 0.90. This implies that the sediment sorption for low and mid-concentration MTAs^V^ mainly occurs on localized sites, where MTAs^V^ forms a monolayer on the surface of natural sediments via a chemisorption mechanism and the migration process is controlled by a second-order rate equation [36,43]. For MTAs^V^ concentration of 0.667 mmol/L, both the above two models obtain a satisfactory value of *R*^2^, but the *R*^2^ value of the pseudo-second-order kinetic model (0.9672) is slightly higher than that of the pseudo-first-order kinetic model (0.9650). This means that a high concentration of MTAs^V^ adsorption may include chemical and physical sorption, but chemisorption is still the major mechanism.

To further determine the major mechanisms affecting the quick, slow, and equilibrium adsorption for MTAs^V^, the elovich, intraparticle diffusion, and fractional power models are also used to fit the experimental data, which are shown in Figure 3 and Table 2. For 0.667 mmol/L MTAs^V^ in the slow and equilibrium adsorption processes (from 50 min to 12 h), a higher value of *R*^2^ obtained using the elovich model (about 0.97) than that obtained using the intraparticle diffusion model (about 0.94) further supports the finding that chemisorption is the major mechanism. Since the elovich and intraparticle models are used for chemisorption and physical sorption systems, respectively [36]. Meanwhile, a high value of *R*^2^ (about 0.97) is also obtained using the fractional power model, indicating that the slow and equilibrium adsorption of sediments for a high concentration of MTAs^V^ occurs on the heterogeneous surface [36]. Similarly, 0.013 mmol/L MTAs^V^ adsorption during the reaction time from 10 min to 12 h also supports the chemisorption mechanism, due to the fact that the fitted *R*^2^ using the elovich and fractional power models is slightly higher than that using the intraparticle diffusion model. In addition, it is found that for 0.667 mmol/L MTAs^V^ adsorption within 50 min, there are nice exponential relationships rather than linear relationships between In(*t*) and *Q_t_*, *t*^0^.^5^ and *Q_t_*, and In(*t*) and In(*Q_t_*) (See Appendix A). This means that the quick adsorption process of high concentration of MTAs^V^ cannot be simply characterized by these three models and this sorption may be a complicated combination of physical and chemical sorption. For 0.002 mmol/L MTAs^V^ in the slow and equilibrium adsorption processes (from 10 min to 12 h), *R*^2^ obtained using the intraparticle diffusion model is greater than 0.95, which is slightly higher than other two models, meaning that there is a stronger possibility of a diffusion rate controlling mechanism than chemisorption mechanism within the system containing low concentration of MTAs^V^ when the sorption is slow or tends to be equilibrium.

### 3.3. Sorption Isotherm of MTAs^V^ by Natural Sediments

Figure 4 shows adsorption isotherm for MTAs^V^ on the sediment and its fitting parameters by the Langmuir, Freundlich, and Langmuir-Freundlich models are listed in Table 3. With the increase of MTAs^V^ equilibrium concentration (*C_e_*), equilibrium adsorption capacities (*Q_e_*) of MTAs^V^ on the sediment increase greatly at first and then tend to be stable at higher concentrations (about more than 0.300 mmol/L). Comparing the coefficients of determination (*R*^2^) obtained by these three models plotting curves, MTAs^V^ adsorptions on the sediment fits the best with the Langmuir-Freundlich model and its *R*^2^ is the highest of 0.9882. This means that at low concentration, MTAs^V^ sorption isotherm by natural sediments becomes the Freundlich isotherm model, while at high concentration of MTAs^V^, its sorption isotherm by natural sediments becomes the Langmuir isotherm model. The Freundlich model assumes that the adsorption is mainly in the form of a multilayer with a heterogeneous surface structure [35]. The Langmuir model assumes the adsorbent surface structure is homogeneous and the adsorption reaction is monolayer adsorption [44,45]. The Langmuir-Freundlich isotherm includes the knowledge of adsorption heterogeneous surface, which describes the distribution of adsorption energy onto the heterogeneous surface of the adsorbent. MTAs^V^ adsorption on natural sediment fits the best into the Langmuir-Freundlich model, indicating that a low concentration of MTAs^V^ adsorption on natural sediment is mainly heterogeneous multilayer adsorption, and the maximum adsorption capacity (*Q_m_*) is mainly determined by the monolayer adsorption for the high concentration of MTAs^V^. That is why *Q_m_* (about 362.22 μg/g) obtained by the Langmuir-Freundlich model fitting is similar to that (about 362.37 ug/g) obtained by the Langmuir model fitting.

### 3.4. Characterization of Sediments before and after MTAs^V^ Sorption

#### 3.4.1. XRD Analysis

Mineral composition results of sediment before and after MTAs^V^ adsorption using XRD analysis (Figure 5) show that the major mineral composition is quartz (SiO_2_) according to the standard card of PDF#46-1045, and the minor mineral compositions are mainly albite (ordered, (Na, Ca)Al(Si, Al)_3_O_8_) and muscovite ((K, Na)(Al, Mg, Fe)_2_(Si_3.1_Al_0.9_)O_10_ (OH)_2_) according to the standard cards of PDF#41-1480 and PDF#07-0042, respectively. After MTAs^V^ adsorption, it is found that all the peak intensities of sediment decrease slightly, indicating that the crystallinity of sediment is weakened. This may be attributed to the complexation of the adsorbed As and Fe-(or Al-) compounds of the sediments, because after MTAs^V^ adsorption, there are almost no changes for the major and minor mineral compositions, while some As-Al and As-Fe compounds are detected on the sediments. Using the As-bearing mineral phase peak match, the adsorbed As compounds on the sediment are probably AlAs, Na_3_Al_2_(AsO_4_)_3_, and Fe_8_As_10_O_23_ according to the standard cards of PDF#17-0915, PDF#51-0346, and PDF#35-0462, respectively. This means that MTAs^V^ adsorption on the sediment is mainly attributed to its complexation with Al- and Fe-bearing compounds.

#### 3.4.2. BET Analysis

Table 4 lists the surface and pore characterization results of the sediment before and after MTAs^V^ adsorption. Before reaction, the single-point surface area, BET surface area, Langmuir surface area, and T-plot external surface area of the sediment are 21.81, 22.44, 34.52, and 20.14 m^2^/g, respectively. All of them increase moderately after MTAs^V^ adsorption. This may be because MTAs^V^ is mainly adsorbed on the surface of the sediment, leading to the surface of sediment becoming rougher. The results of kinetic and isotherm adsorption (Section 3.2 and Section 3.3) and the changes of surface morphology of sediment after adsorption can further confirm this conclusion (Figure 6). Furthermore, the mean value of pore volume of sediment decreases slightly from 6.43 cm^3^/g to 6.15 cm^3^/g (Table 4), which may be attributed to the fact that the adsorbed As compounds also partially fill the pores of the sediment.

#### 3.4.3. SEM-EDS Analysis

The surface morphology of sediment before and after MTAs^V^ adsorption is determined using the Scanning Electron Microscope (SEM), and the results are shown in Figure 6. Meanwhile, elemental analysis of sediment is determined using the Energy Dispersion Spectrum (EDS) and the results are listed in Table 5. Before adsorption, the surface structure of the sample is smooth, while some small crumps and crystals occur on the surface after adsorption. EDS results show that before MTAs^V^ adsorption, the major elements of sediment and their average percentages are in the order of O (39.80%), Si (28.68%), Fe (13.69%), and Al (12.28%). After adsorption, on the sites where no crumps and crystals are found on the surface of the sediment, none of As is detected and the major elements and their average percentage are in the order of O (46.85%), Si (24.58%), Fe (8.44%), Mg (7.34%), and Al (6.99%). However, on the sites with some crumps and crystals, As is detected at the range of 0.14–0.46% and the major elements and their average percentages are in the order of O (32.05%), Si (31.62%), Fe (19.09%), and Al (14.28%). This means that MTAs^V^ adsorption mainly occurs on the surface sites with higher contents of Fe and Al. In addition, none of S is detected on the surface of sediments after MTAs^V^ adsorption, indicating that the adsorbed MTAs^V^ probably transforms into other As species, which are probably AlAs, Al-As-O, and Fe-As-O compounds according to the former XRD analysis results (Section 3.4.1).

### 3.5. Mutual Interferences between MTAs^V^ and Other as Species Sorption

Figure 7 shows the changes of MTAs^V^, arsenite, and arsenate sorption capacities (*Q_t_*) with time in natural sediments and groundwater systems containing single As ion and binary As species, respectively. It can be seen that MTAs^V^, arsenite, and arsenate adsorption kinetics show similar trends both in single As ion and binary As ions systems. In which, arsenate shows the highest equilibrium sorption capacity (*Q_e_*) to natural sediments, which is about twice as large as that of MTAs^V^. Previous studies have found that amorphous Al(OH)_3_ and α-FeOOH can also adsorb much more arsenate compared to MTAs^V^ [27,28,31]. This phenomenon may be attributed to the fact that the SH group of MTAs^V^ has a larger volume than the OH group of arsenate, inhibiting the surface complexation of MTAs^V^ with sediments [27,31]. Arsenite shows a slightly lower sorption capacity than MTAs^V^ and a much lower sorption capacity than arsenate to natural sediments. This is probably because the major compositions of natural sediments including Fe- and Al-bearing minerals are responsible for As adsorption from aqueous solutions. It has been reported that both Al_2_O_3_ and Fe_2_O_3_ show higher adsorption capacities for arsenate compared to arsenite [46,47]. On one hand, amorphous Al oxide has a higher site saturation for arsenate (3.8 × 10^−2^ μmol/m^2^) than arsenite (2.3 × 10^−2^ μmol/m^2^) [46]. On the other hand, at a pH of 7, the positively charged Fe_2_O_3_ has a stronger electrostatic attraction to the negatively charged arsenate than the electrically neutral arsenite, therefore, it can adsorb more arsenate (*Q_e_* = 4122 ± 62.79 μg/g) than arsenite (*Q_e_* = 2899 ± 71.09 μg/g) [47].

Comparing MTAs^V^ adsorption in single As ion and binary As ions systems (Figure 7a), it can be seen that the added arsenite and arsenate greatly decreases MTAs^V^ sorption, and the addition of arsenate causes more decrease of MTAs^V^ adsorption capacities (*Q_t_*) than arsenite, indicating that there is a significant competitive effect of the added arsenite and arsenate on the adsorption of MTAs^V^ on natural sediments, and arsenate can compete more favorably than arsenite for MTAs^V^ sorption. Furthermore, the presence of MTAs^V^ also causes a slight reduction of arsenite adsorption on sediments before 4 h, but this effect becomes weaker when approaching the equilibrium state (Figure 7b). It is found that in the binary ions system containing MTAs^V^ and arsenite, the experimental *Q_e_* of MTAs^V^ greatly decreases from 216.47 μg/g to 169.98 μg/g and the experimental *Q_e_* of arsenite decreases from 140.03 μg/g to 131.88 μg/g compared to the single ion system. Since MTAs^V^ is stable and has no species change before 24 h in this study (Section 3.1), the observed competitive effect between MTAs^V^ and arsenite adsorption on natural sediments may be attributed to both these two As species can complex with Al- and Fe- oxyhydroxides [27,31]. The difference in mutual competition intensity may be caused by the complexation degree of Al- and Fe- oxyhydroxides with MTAs^V^ and arsenite being different.

Different from arsenite, the added arsenate greatly decreases MTAs^V^ sorption, and the presence of MTAs^V^ also greatly decreases arsenate sorption in the binary As ions system (Figure 7a,c). In the single ion system, the experimental equilibrium sorption capacities (*Q_e_*) of MTAs^V^ and arsenate on natural sediments are 216.47 μg/g and 460.25 μg/g, respectively. However, in the binary ions system containing MTAs^V^ and arsenate, the experimental *Q_e_* values of MTAs^V^ and arsenate are reduced by about 40% and 26%, respectively, and decrease to 130.06 μg/g and 341.67 μg/g, respectively. The strong competition between arsenate and MTAs^V^ may be attributed to the fact that these two As species show a similar dissociation constant and chemistry structure [31,48,49] leading to the fact that these two As species show similar complexation with Al- and Fe- minerals, and they can occupy the sorption sites on the surface of sediments to the same extent with each other. Due to the limited understanding of thioarsenate at present, this is the first observation of a competitive effect between MTAs^V^ and arsenate adsorption on the sediment.

## 4. Conclusions

MTAs^V^ is one of the major thioarsenic species in natural sulfidic waters and is an important intermediate product during thioarsenic transformation to arsenite or arsenate, thus it is critical to understand its sorption to natural sediments and the influence of other As species on MTAs^V^ sorption. In this study, batch experiments are carried out to investigate the adsorption characteristics and mechanisms of MTAs^V^ sorption to natural sediments, and to identify the effects of arsenite and arsenate on MTAs^V^ sorption. The major conclusions are as follows:

For the low, medium, and high concentrations of MTAs^V^ (0.002, 0.013, and 0.667 mmol/L, respectively), their adsorption by the natural sediments all increase rapidly in a short time and then increase slowly before 8 h, and finally approaches an asymptote after that.Chemisorption is the major mechanism of MTAs^V^ sorption to the sediment because MTAs^V^ adsorption experimental data fit better into the pseudo-second-order kinetic model with a coefficient of determination (*R*^2^) greater than 0.90. At low concentration, MTAs^V^ sorption isotherm by natural sediments becomes the Freundlich isotherm model, while at high concentration of MTAs^V^, its sorption isotherm by natural sediments becomes the Langmuir isotherm model.MTAs^V^ adsorption on natural sediments is mainly attributed to its complexation with Al- and Fe-bearing compounds. Since characterization results show that the sediment sorption for MTAs^V^ mainly occurs on the localized sites with higher contents of Fe and Al, where the adsorbed MTAs^V^ probably transforms into other As species, such as AlAs, Al-As-O, and Fe-As-O compounds. These produced compounds may increase the surface area and decrease the pore volume of sediment.There is a competitive effect in MTAs^V^ and arsenite adsorption, and in MTAs^V^ and arsenate adsorption on natural sediments. The presence of arsenite greatly decreases MTAs^V^ sorption, while the presence of MTAs^V^ causes a certain degree of reduction of arsenite adsorption on natural sediments before 4 h and this effect becomes weaker when approaching the equilibrium state. The presence of arsenate greatly decreases MTAs^V^ sorption and the presence of MTAs^V^ also greatly decreases arsenate sorption. The different competition effects between MTAs^V^ and arsenite may be caused by the different intensities of their complexation with Al- and Fe-bearing compounds. On the contrary, the similar competition effects between MTAs^V^ and arsenate can be attributed to their similar structure and similar complexation with Al- and Fe-bearing compounds.

## Figures and Tables

**Figure 1 ijerph-18-12839-f001:**
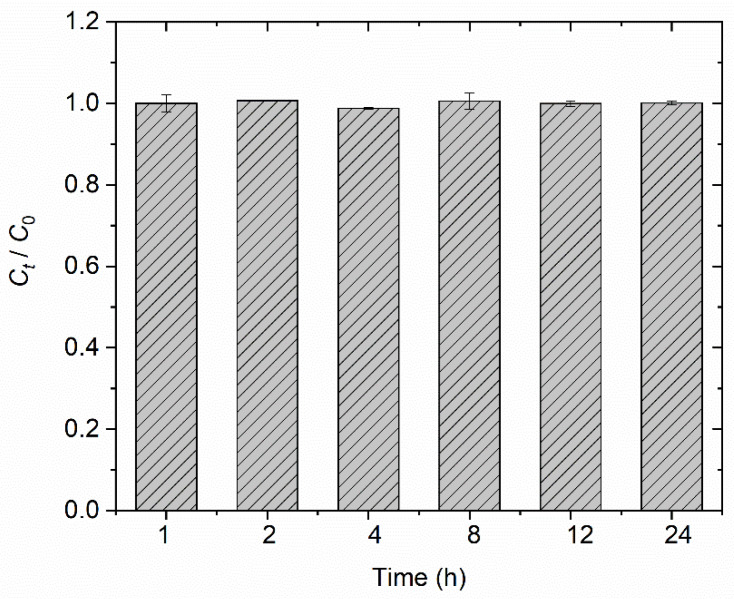
The changes of *C_t_*/*C*_0_ for MTAs^V^ solutions over time in the blank experiment.

**Figure 2 ijerph-18-12839-f002:**
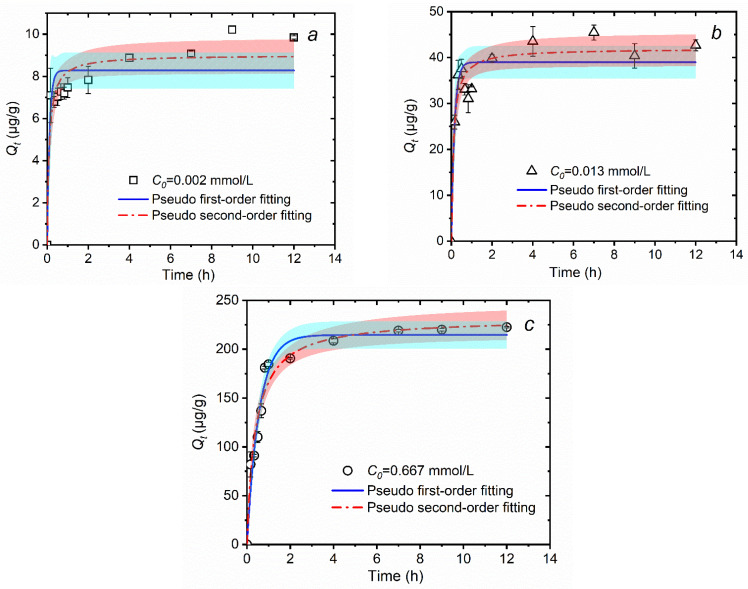
Kinetic adsorption of MTAs^V^ on the natural sediment and their fitting curves using pseudo-first-order and pseudo-second-order models, in which the initial concentrations of MTAs^V^ (*C*_0_) are 0.002 (**a**), 0.013 (**b**), and 0.667 (**c**) mmol/L, respectively. The shadow represents the confidence band under the 95% confidence level for fitting curves.

**Figure 3 ijerph-18-12839-f003:**
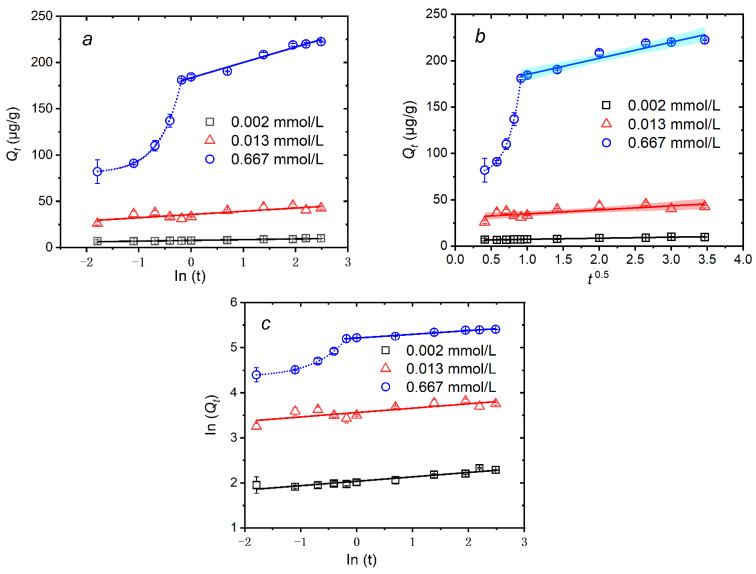
Fitting curves of MTAs^V^ adsorption on the natural sediment using elovich (**a**), intraparticle diffusion (**b**), and fractional power (**c**) models, respectively. The shadow represents the confidence band under the 95% confidence level for fitting curves.

**Figure 4 ijerph-18-12839-f004:**
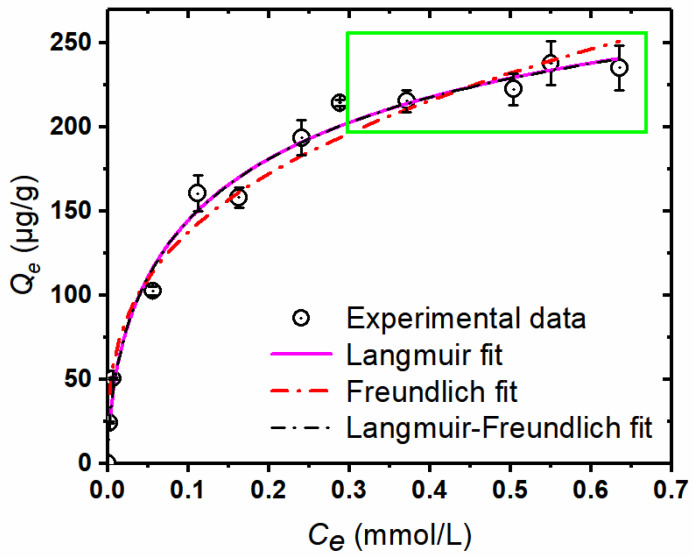
Adsorption isotherms of MTAs^V^ on 0.50 g/50 mL sediments at pH 7 and 25 °C. The data in the green rectangle represent a stable trend of MTAs^V^ sorption (*Q_e_*) at higher As equilibrium concentrations (*C_e_*).

**Figure 5 ijerph-18-12839-f005:**
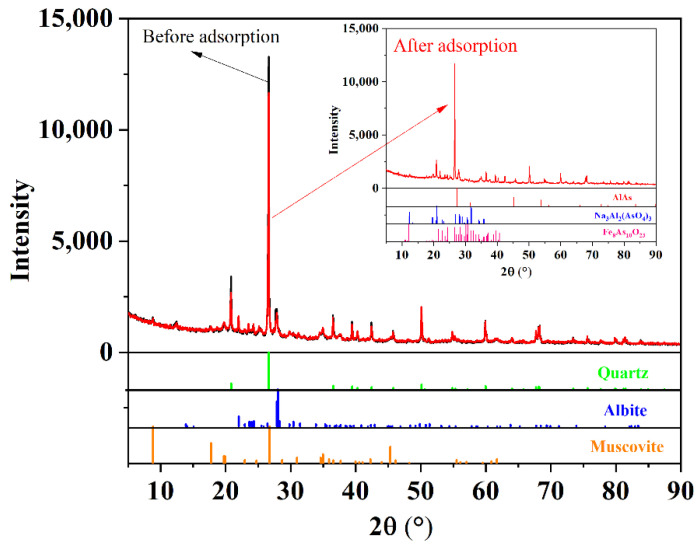
XRD patterns and the mineral compositions analysis of sediment samples before and after MTAs^V^ adsorption.

**Figure 6 ijerph-18-12839-f006:**
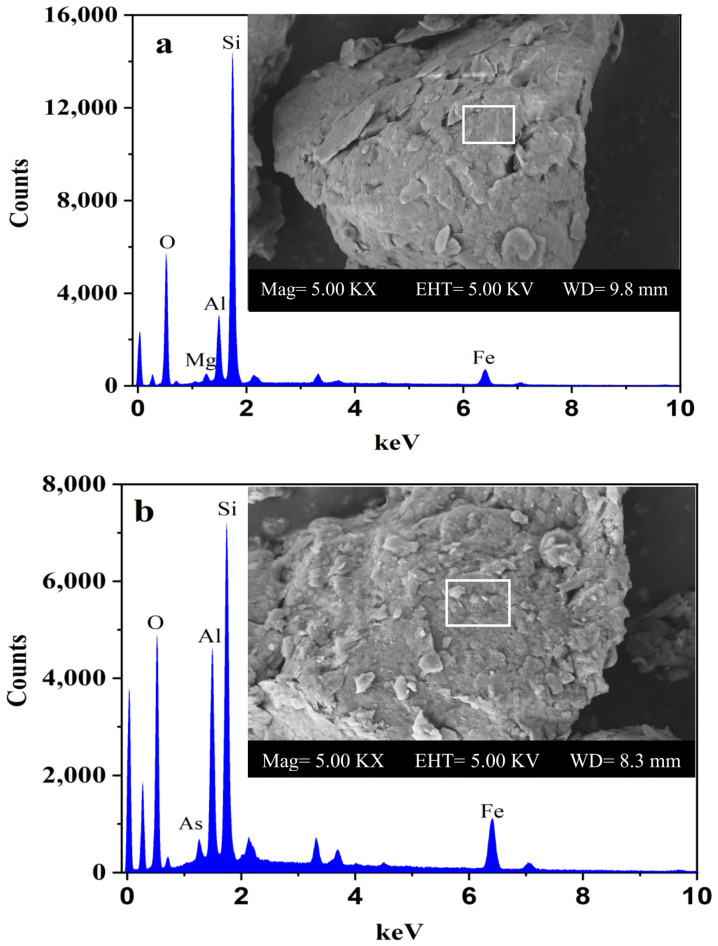
Scanning electron microscope (SEM) photos and energy dispersion spectrum (EDS) of sediment samples before (**a**) and after (**b**) MTAs^V^ adsorption.

**Figure 7 ijerph-18-12839-f007:**
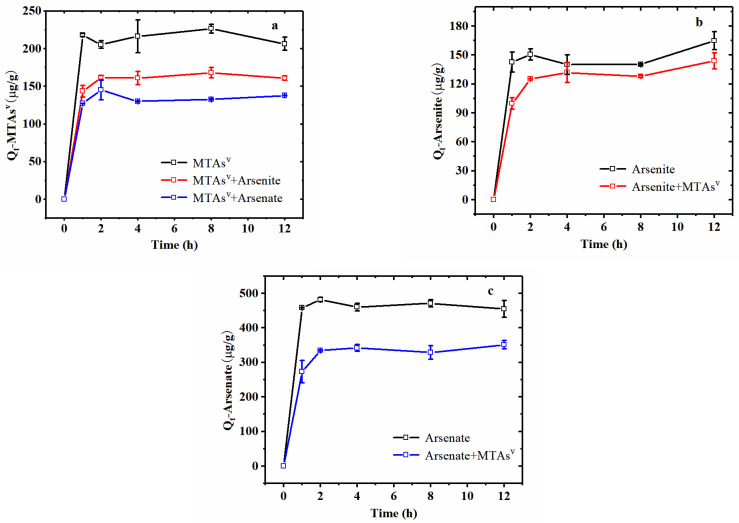
Kinetic adsorption of MTAs^V^ (**a**), arsenite (**b**), and arsenate (**c**) in single ion system and binary ions system, in which the initial concentrations of MTAs^V^, arsenite, and arsenate are 0.627 mmol/L.

**Table 1 ijerph-18-12839-t001:** Hydrochemical characteristic of synthesized groundwater (SGW) solutions.

Parameters	pH	ORP	EC	Cl^−^	NO_3_^−^	SO_4_^2−^
Units		mv	µs/cm	mg/L		
Value	7	−79	289	3.6	6.4	16.2
**Parameters**	**HCO_3_^−^**	**K^+^**	**Ca^2+^**	**Na^+^**	**Mg^2+^**	
Units	mg/L					
Value	140.949	8.424	90.68	2.804	0.765	

**Table 2 ijerph-18-12839-t002:** Parameters of five kinetic models for MTAs^V^ adsorption on the sediments.

*C*_0_ (mmol/L)	Reaction Time	Pseudo-First-Order				Pseudo-Second-Order				
		*Q_e_*	*k* _1_	*R* ^2^		*Q_e_*	*k* _2_		*R* ^2^	
0.002	0 ≤ t ≤ 12 h	8.280 ± 0.386	8.756 ± 3.489	0.8238		9.009 ± 0.386	1.151 ± 0.442		0.9049	
0.013		38.978 ± 1.599	6.430 ± 1.793	0.8739		41.935 ± 1.643	0.225 ± 0.075		0.9228	
0.667		214.557 ± 6.346	1.766 ± 0.177	0.9650		231.799 ± 7.746	0.010 ± 0.001		0.9672	
** *C* _0_ ** **(mmol/L)**	**Reaction Time**	**Elovich**			**Intraparticle Diffusion**			**Fractional Power**		
		**1*/β***	**ln*(αβ*)/*β***	** *R* ^2^ **	** *k_i_* **	** *c* **	** *R* ^2^ **	** *b* **	**In (*a*)**	** *R* ^2^ **
0.002	10 min ≤ t ≤ 12 h	0.803 ± 0.096	7.730 ± 0.136	0.8861	1.115 ± 0.083	6.343 ± 0.154	0.9517	0.097 ± 0.010	2.035 ± 0.014	0.9081
0.013		3.486 ± 0.759	35.716 ± 1.080	0.7008	4.326 ± 1.172	30.489 ± 2.164	0.6022	0.097 ± 0.022	3.561 ± 0.031	0.6818
0.667	50 min ≤ t ≤ 12 h	16.636 ± 1.092	183.453 ± 1.713	0.9789	17.382 ± 1.751	167.864 ± 4.189	0.9463	0.082 ± 0.005	5.212 ± 0.008	0.9787

**Table 3 ijerph-18-12839-t003:** Isotherm parameters for adsorption of MTAs^V^ on the sediment.

Ce (mmol/L)	Models Fitting			
0–0.635	Langmuir fitting			
	*Q_m_* (μg/g)	*K_L_*		*R* ^2^
	362.37 ± 70.58	2.58 ± 1.48		0.9859
	Freundlich fitting			
	*K_F_*	1/*n*		*R* ^2^
	290.75 ± 11.53	0.32 ± 0.02		0.9698
	Langmuir-Freundlich fitting			
	*Q_m_* (μg/g)	*K_LF_*	*m*	*R* ^2^
	362.22 ± 66.68	4.994 ± 3.37	0.59 ± 0.09	0.9881

**Table 4 ijerph-18-12839-t004:** Characterization parameters of sediment before and after MTA adsorption.

Characterization	Parameters	Before Reaction	After Reaction
Surface area (m^2^/g)	Single point surface area	21.81	23.09
	BET surface area	22.44	24.48
	Langmuir surface area	34.52	37.74
	T-plot external surface area	20.14	24.33
Pore volume (cm^3^/g)	Mean value	6.43	6.15

**Table 5 ijerph-18-12839-t005:** Elemental analysis results of sediment before and after MTAs^V^ adsorption.

Elements	Before Adsorption (wt%)			After Adsorption *^a^* (wt%)			After Adsorption *^b^* (wt%)		
	Min	Max	Average	Min	Max	Average	Min	Max	Average
O	31.26	47.77	39.80	45.59	48.60	46.85	25.33	37.94	32.05
Si	18.78	42.51	28.68	21.60	26.84	24.58	26.18	35.56	31.62
Fe	13.00	14.33	13.69	5.04	13.59	8.44	14.84	24.99	19.09
Al	7.92	15.46	12.28	1.58	11.08	6.99	8.27	17.46	14.28
K	1.17	2.71	2.02	0.22	1.69	1.09	0	3.28	1.29
Mg	0.99	3.64	2.44	3.68	11.39	7.34	0	1.59	1.13
Ca	0.58	0.70	0.64	1.00	7.95	4.55	0	0.48	0.12
Na	0	0.37	0.20	0	0.22	0.06	0	0	0
As	/	/	/	/	/	/	0.14	0.46	0.30

*^a^*: Sediment sample that none of As is detected on the surface after MTAs^V^ adsorption. *^b^*: Sediment sample that As is detected on the surface after MTAs^V^ adsorption.

## Data Availability

All data used during the study appear in the submitted article.

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
