# Peer review of "Sorption of Monothioarsenate to the Natural Sediments and Its Competition with Arsenite and Arsenate"

_ijerph, 2021, doi:10.3390/ijerph182312839_

Round 1
Reviewer 1 Report
The manuscript “ijerph-1466992” entitled “Sorption of monothioarsenate to the natural sediments and its competition with arsenite and arsenate” deals with the sorption properties of natural sediments towards monothioarsenate (MTAsV). The manuscript is seriously lacking in scientific aspects. The base of research design is inappropriate and the authors have rarely discussed the significance and scientific contribution of current study. Following are my comments which will further suggest reconsideration of manuscript preparation to the authors in terms of its appropriate design, research objectives, presentation of results etc.
- First of all, the authors are claiming that they are conducting study on the sorption of MTAsV on natural sediments. One of the biggest scientific flaws lies in the composition of natural sediments. Since the authors are claiming that they collected natural sediments from area having severe groundwater contamination with arsenic (As) species. If this is so, the natural sediments of that area must contain As content as well. How comes the researchers analyze natural sediments with SEM-EDX, XRD and still none of the analysis confirm the presence of As in soil sediments. It is impossible for any area containing huge concentration of As in groundwater sample with no As content in that soil. This is something which indicates that authors didn’t conducted research properly. Further adding, when I look at the SEM-EDX and XRD analysis results, none of the graph clearly indicate the difference between before and after sorption analysis. It clearly indicates both XRD and SEM-EDX are not done accurately by researchers.
- Second most important point regarding the results and discussion section of current manuscript is that the authors have prepared manuscript in a very tidy manner. For instance, they don’t know how to present results of for example kinetics experiments. They are showing nonlinear as well as linear fitting parameters in the figure and main text but now one can pick up that what researchers are exactly talking about. There is a lot of loose discussion without setting any baseline. For example, in the line 168-192, the researchers have discussed all possible aspects of kinetic models but they didn’t come up with any thorough conclusion. They are claiming Pseudo-second order is best fit and at the same time they are saying that there is a role of intra particle diffusion. It seems they themselves are confused exactly what their experimental data best fits.
- Similarly, there is no proper conclusion on adsorption isotherm fitting. The researchers claim both Langmuir and Freundlich best fits their experimental data. How is that true? The researchers can focus on modified Langmuir-Freundlich isotherm model for the proper answer of this question.
- In addition, the researchers are totally mixing the methodology and results discussion section. For instance, the assumptions, mathematical expressions of kinetics/isotherms model should be moved to methodology section. Another most important point is that if researchers are using non linear model for fitting their experimental data, they should include mathematical expression of none linear model. There exist severe mistakes in the manuscript.
- Another query regarding SEM analysis is that the researchers just show the SEM images with incomplete detail regarding its resolution plus no discussion is found in the main text of the manuscript.
- Further adding, why decrease in pore density resulted in increase in surface area? Again, no discussion by researchers on this main point as well.
- One of the key question arise on the competitive removal of MTAsV with As(III) and As(V), the graphs are generally the representation and pictorial expression of how different species behave under competitive environment. No one can pick it easily from graph. As well as the discussion must present reasons about what is happening and why is it so?
- The researchers said MTAsV is present in water. There is a valid question regarding its toxicity, concentration ranges in aquatic environment of world etc. The researchers didn’t clarify it.
- Line 109! Is it 2328 mg/L It’s almost impossible for any area of the world to present such a huge contamination of As species?
- The researchers have rarely used recent references. Only 9 references are from last 5 year out of 44. This makes another question mark on the suitability of manuscript in the current form.
- In addition, do check English of your manuscript. I think there is sever lacking of scientific as well as grammatical English in the current manuscript.
Reviewer 2 Report
In my opinion, authors should refine the paper, so that it can be understood or explain more some details. The novelty provided by the article is not justified, in view of the references provided. Some changes should be done, as for example:
- Section 2.2, 1st paragraph: There are two sentences that are almost equal. Please revise from line 121 to line 127
- Table 1, 2nd column: HCO3 the minus must be super index
- Hour is not an unit of the International System although it is allowed, but the symbol is h. Please, replace hr with h (for more information: https://www.bipm.org/en/home)
- For the sorption isotherms Which is the temperature? Is it the same for all measurements?
- Table 2.- There are not results for the first 10 min in the last 3 models in this table, neither in the figure 3. But, for example, in line 234 you said interparticle diffusion model fits well for data of 0.013 mmol/L. We cannot verify this claim
- Figure 3.- Can you explain the behaviour of the first five points for higher concentration?
- BET analysis: The BET area increase cannot be attribute to the decrease of pore size, if the number of pore is not change (lines 309-311)
- Line 320: how do you know if the surface is smooth? Have you measure de roughness?
Round 2
Reviewer 1 Report
The authors have significantly improved the manuscript as per my suggestion. As a minor comment, please change the Pseudo first order and second-order equation from its linear form to non-linear form in the methodology section. After this correction, the manuscript can be accepted for publication.
Reviewer 2 Report
Paper has been improved properly.
Author Response
Thanks for the reviewer’s suggestion!